# Building Research Infrastructure: The Development of a Technical Assistance Group-Service Center at an RCMI

**DOI:** 10.3390/ijerph20010191

**Published:** 2022-12-23

**Authors:** Monica R. Lininger, Christine Kirby, Kelly A. Laurila, Indrakshi Roy, Marcelle Coder, Catherine R. Propper, Robert T. Trotter, Julie A. Baldwin

**Affiliations:** 1Department of Physical Therapy and Athletic Training, Northern Arizona University, Flagstaff, AZ 86011, USA; 2Southwest Health Equity Research Collaborative, Northern Arizona University, Flagstaff, AZ 86011, USA; 3Center for Health Equity Research, Northern Arizona University, Flagstaff, AZ 86011, USA; 4Department of Biological Sciences, Northern Arizona University, Flagstaff, AZ 86011, USA

**Keywords:** research infrastructure, REDCap, early stage investigator, methodological support, behavioral and social sciences

## Abstract

As one of the Research Centers for Minority Institutions (RCMI), the Southwest Health Equity Research Collaborative (SHERC) worked over the first five-year period of funding to foster the advancement of Early Stage Investigators, enhance the quality of health disparities research, and increase institution research capacity in basic Biomedical, Behavioral, and/or Clinical research; all priorities of RCMIs. In year 4, the Technical Assistance Group-Service Center (TAG-SC) was created to help achieve these goals. The TAG-SC provides one-on-one investigator project development support, including research design, data capture, and analysis. Successful implementation of the TAG-SC was tracked using Research Electronic Data Capture (REDCap), a secure, web-based software platform allowing for immediate tracking and evaluation processes. In the first two years, 86 tickets were submitted through the REDCap system for methodological support by TAG-SC experts (faculty and staff) for assistance with health-equity related research, primarily SHERC and externally funded Social/Behavioral research projects. The TAG-SC increased the research capacity for investigators, especially within the SHERC. In this manuscript, we describe the methods used to create the TAG-SC and the REDCap tracking system and lessons learned, which can help other RCMIs interested in creating a similar service center offering an innovative way to build methodological infrastructure.

## 1. Background

In September 2016, the Center for Health Equity Research (CHER) [1] was formally established to collaborate with communities to build foundations and environments that support health and wellbeing in the Southwest United States. CHER houses the Southwest Health Equity Research Collaborative (SHERC) [2], which was funded (U54MD012388) in 2017 via the National Institutes of Health, National Institute on Minority Health and Health Disparities (NIMHD) [3] through the Research Centers in Minority Institutions (RCMI) Program [4]. The goals of the RCMI Program Specialized Centers [5] include: (1) fostering environments conducive to career enhancement with a particular emphasis on the development of new and early career investigators; (2) enhancing the quality of all scientific inquiry and promoting research on minority health and health disparities; and (3) enhancing institutional research capacity within the areas of basic Biomedical, Behavioral, and/or Clinical research. The RCMI program was created in 1985 and has significantly increased knowledge related to health disparities science while simultaneously expanding the diversity of biomedical research [6].

The Hispanic-Serving designation and the institution’s strategic commitment to Indigenous Peoples uniquely position SHERC to engage underrepresented (UR) communities in research, advance culturally competent translational science and improve health outcomes for underserved communities (RFA-MD-17-003). Within SHERC, as with all RCMIs, there are five cores, each with a different objective to achieve the overall center mission. One of the five cores within SHERC is the Research Infrastructure Core (RIC), whose responsibility, in part, is to develop faculty-level expertise in research design and advanced methodology (including biostatistics, team science, bioinformatics, data science, qualitative methods, and health informatics). Additionally, the core is charged with linking this group of experts with investigators at the institution to increase research capacity and infrastructure [7,8].

One of the ways to meet these goals is to construct a service center composed of faculty and staff with expertise in various research designs, methodologies, and analyses to enhance the institution’s capacity to conduct cutting-edge health research. Since 2017, SHERC has been working towards the genesis of this service center. Therefore, the purpose of this manuscript is to describe the critical processes and initial outcomes in the development of a Technical Assistance Group-Service Center (TAG-SC) at a southwestern RCMI, including the development of a ticketing system through the Research Electronic Data Capture (REDCap) [9,10], a secure, web-based software platform.

## 2. Methods

### 2.1. Technical Assistance Group-Service Center (TAG-SC) Development

In the following sections, we will describe the creation of the TAG-SC in three phases to highlight its progression over the first grant cycle of SHERC (2017–2022).

#### 2.1.1. Phase I: Initial Framework (Grant Years 1–3, 2017–2020)

In the initial years (2017–2020), the RIC team started with 14 faculty/staff, including a core lead, co-lead, coordinator, and 11 faculty, to assist with requests. Researchers would contact the RIC lead to request consultations to support a specific methodological or statistical aspect of their health research. Based on need, the RIC lead assigned each request to a RIC faculty/staff member with expertise in the particular area. The RIC faculty/staff would work with the researcher to accomplish the requested level of support. RIC faculty were compensated with a one-course release from teaching for their assistance during the academic year. Staff assistance was provided by assigning specific projects or activities as a part of regular staff duties as named personnel on the grant. Additionally, during this phase of development, RIC faculty/staff were asked to report consultations in an Excel spreadsheet which included the type of assistance provided [proposal development, proposal review, consultation (not proposal related), laboratory services, or biostatistical support] and document key RCMI measures including university department, type of project (Biomedical, Clinical, Social/Behavioral), whether the project was health equity related, outcomes (i.e., manuscripts or grants submitted) and hours spent on the consultation. Through these early years, we successfully determined the needs of researchers across campus (via environmental scans starting in 2019), found RIC faculty/staff with varied expertise to meet these needs, and implemented tracked progress and outcome evaluation.

##### Developing a Ticket System Using REDCap

In the first three years, it became apparent that a modified ticket tracking system was needed to improve the efficiency of assistance workflow and decrease recall bias by the RIC faculty/staff members. We transitioned from manual tracking to a ticket system housed within REDCap, a secure web platform for constructing and managing online databases and surveys. [11] Using REDCap allowed for a more streamlined process for the researcher and the RIC faculty/staff member providing support. It also provided a database to refine targeted time and need-based allocation of limited resources.

The TAG-SC REDCap ticketing system was intended to satisfy multiple stakeholders’ needs, including administrative requirements (tracking faculty/staff support for compensation), RIC team project management (documenting team assignments, approval of requests, tracking/reporting essential functions), and process and outcome evaluation of SHERC funded activities. The development of the initial REDCap project spanned months of investigation and research. The shell building, testing, and brainstorming with direct participation from the Research Infrastructure and Administrative Cores (including the evaluation team) took 76 days, determined by the time-in-development feature in REDCap.

The REDCap TAG-SC project architecture is comprised of four components; public facing, internal team processes, feedback, and evaluation. As depicted in Figure 1, a support request gets filled out via a survey link on the TAG-SC webpage hosted by the University. This request support form consists of requestor information and additional details regarding necessary assistance. Table 1 highlights some of the questions in the “request support” form. Once submitted, the TAG-SC director receives an automated alert via email. To begin the internal team process, the director logs into REDCap to review the request, approves or denies assistance, logs administrative data, and assigns the ticket to a TAG-SC methodological expert (faculty or staff). The internal process is only accessible to members of the TAG-SC team via REDCap User Rights. The director notifies the requestor and methodological expert of the status of the request and relevant details. Then, the methodological expert and requestor work together to achieve the intended goal of the request (instrument development, software training, quantitative analysis, qualitative methodologies, etc.). To conclude the internal process, the methodological expert logs the details of completing each ticket (hours, notes on service provided, etc.) in the conclusion and tracking form. Once their approved goal or time limit is met, the methodological expert “closes” the ticket in REDCap; each ticket is closed following assistance completion. Upon closure, a brief satisfaction survey is deployed internally called a “Yelp” review. This survey includes three items measuring if the investigator’s needs were met by the service provided, the overall recommendation of the TAG-SC, and finally, an open-ended item for general feedback. The outcome survey includes outcome oriented questions to document measurable products such as proposal submissions, manuscript development and publication, new research collaborations, conference presentations, and data use agreements.

Many lessons were learned during the initial development phase, such as revising the compensation policy and procedures for RIC faculty/staff, evolving researcher requests for targeted methods, and conducting evaluation follow-up after assistance was provided. During Phase I: Initial Framework, RIC faculty were compensated through academic course release. We subsequently found that the number of requests for specific faculty assistance and time spent with each request was unequal across faculty; consequently, the single course release per faculty member was not equitable, cost-effective, or compatible with changing administrative policy.

#### 2.1.2. Phase II: Pilot Year (Grant Year 4, 2020–2021)

Building on the framework developed during phase I, we transitioned from tracking assistance in Excel to REDCap, increasing reporting accuracy and efficiency. In each Phase, we conducted periodic “environmental scans” (university wide methods-needs surveys) to inform each development process. In the early years, the type of assistance requested focused predominately on traditional (qualitative and quantitative) methodologies and analyses; however, the results from subsequent environmental scans demonstrated that RIC expertise needed to be expanded to include mixed-methodologies, machine learning, geographical information systems, and specific software needs (REDCap, NVivo, R, etc.). The RIC leadership team engaged additional university faculty and staff with expertise in these areas to support requests for these types of consultation, which was made possible by the change to an hourly based compensation system in place of the original course release system that involved a much smaller faculty resource group. The combination of expanded expertise and targeted compensation allowed us to pilot-test both the structure and output/impact of a potentially self-sustaining technical assistance program.

#### 2.1.3. Phase III: Year 1 of TAG-SC (Grant Year 5, 2021–2022)

Minor modifications were made to the REDCap project during year 1 of the TAG-SC to document important evaluation metrics, specifically asking investigators about the number of years since the completion of their highest terminal degree (to track Early Stage Investigator [ESI] support) and whether the work was interdisciplinary. The most significant structural change was the TAG-SC transition into a stand-alone service center in CHER rather than being housed in and dependent on SHERC. This change was made to achieve a self-sustaining model over time. A director role was also created with responsibilities of ticket assignment, marketing, faculty/staff recruitment, partnership with financial advisors for all billing, and ultimate reporting of outcomes in collaboration with the evaluation team. The TAG-SC team of specialists increased by two additional faculty/staff members to 16, including the director role. These additional faculty members had research agendas focused on ‘big-data’ analyses and geographic information systems, areas of expertise lacking within the TAG-SC.

To formalize the TAG-SC as a university-sanctioned service center, the TAG-SC director and CHER assistant director worked closely with the university comptroller’s office to create a business plan and formalize rate-setting. The TAG-SC business plan set out the goals and purpose of the service center, staffing and services to be offered, phased goals with benchmarks, a marketing plan, and a financial sustainability plan.

Rate-setting for the TAG-SC involved calculation of actual hourly costs for service providers, estimated administrative time and associated costs, estimates of the volume (in hours) of each type of offered service for an entire year, and calculation of a 60-day reserve balance per comptroller policy. After the first year of operation (August 2021–June 2022), existing rates were evaluated and modified, and rates were established for new services.

The TAG-SC was able to provide the following types of research assistance during Phase III: a qualitative or quantitative consultation (2 h maximum), qualitative or quantitative analyses (10 h maximum), mixed-methodological analytic support (10 h maximum), instrument development (2–10 h depending of the level of support requested), REDCap assistance (2–10 h depending of the level of support requested), and proposal review (2–10 h depending of the level of support requested).

The TAG-SC director also worked closely with the SHERC communications director to produce a new website (https://nau.edu/cher/cher-tag-service-center/ (accessed on 10 September 2022)) that concentrated on the types of assistance provided to investigators, how to submit a ticket request and brief testimonies from previous users of the TAG-SC. The main goal was to generate a user-friendly website that increased accessibility for the end user. Additional marketing materials, such as flyers, including quick response (QR) codes and resources highlighting available support, were created and distributed to university faculty and staff at various events (e.g., new faculty orientation, departmental meetings, writing manuscript workshop, etc.).

### 2.2. Statistical Analyses

From the REDCap system, we collected data specific to ESI status, professional role at the university, request for assistance, the research content area, type of project, and project details. During Phase II: Pilot Year and Phase III: Year 1 of TAG-SC, three slightly different REDCap projects were created for investigator requests. Minimal modifications were made between each phase to increase the user-friendliness of the request form, better align it with billing processes, and shared data elements across all SHERC cores. For this manuscript, we used six distinct variables with overlapping response options (Table 2) to assess only closed tickets during these two phases of the TAG-SC development. Using these six variables with common response options, we calculated frequencies and percentages across all three REDCap projects encompassing Phase II (2020–2021) and Phase III (2021–2022).

## 3. Results

### 3.1. Phase I: Initial Framework (Grant Years 1–3, 2017–2020)

The team supported fifty-three tickets during the initial three-year timeframe of SHERC. The main distinctions between questions in the REDCap system (phase II) and the initial framework (phase I) included adding questions ascertaining professional roles and ESI status. More than half (64%, 34/53) of tickets directly supported proposal development, while 36% (19/53) of consultations were not proposal related. Almost half (40%, 21/53) of the cases involved SHERC funded pilot or research projects, 26% (14/53) supported external grants, 11% (6/53) were internal grants, and 23% (12/53) were reported as other (non-specified). Biostatistics related requests represented 34% (18/53) of cases. Notably, most projects (89%, 47/53) were health equity related. Requestors could report multiple options regarding the RCMI research foci; therefore, totals are not equivalent to individual tickets (N = 53). A majority (64%, 34/53) of tickets supported Social/Behavioral projects, followed by Biomedical (32%, 17/53), Clinical (28%, 15/53), and missing records accounted for 8% (4/53) cases.

### 3.2. Phase II: Pilot Year (Grant Year 4, 2020–2021) and Phase III: Year 1 of TAG-SC (Grant Year 5, 2021–2022)

Eighty-six tickets were submitted and closed during this two-year timeframe. Nearly 21% (18/86) of requestors were of ESI status (Figure 2), although it should be noted that almost 34% did not report the date of terminal degree; therefore, an ESI designation could not be established for this group. Figure 3 shows that most were faculty members (66%, 57/86) or staff (16%, 14/86). The vast majority of requests centered on either funded projects (31%, 27/86) or proposal development (34%, 29/86) (Figure 4). SHERC funded research projects comprised 21% (18/86) of tickets, while 15% (13/86) supported externally funded grants (Figure 5). Requestors could select multiple options regarding the content area for the research; therefore, totals are not based on single tickets (N = 86). Of the closed ticket requests, over half (65%, 56/86) included an element of Social/Behavioral research (Figure 6). The second most common type of research was Clinical (41%, 35/86), and Biomedical requests only accounted for 19% (16/86). We offer two examples to contextualize how UR ESI faculty and postdocs are supported through the TAG-SC, and the implications that support has (directly/indirectly) in supporting their research career advancement. One requestor received quantitative methodological/analytical support (a power analysis plan) on a social/behavioral focused NIH K-mechanism proposal. This resulted in the first K-award in the history of NAU (awarded to a Native American post-doctoral scholar). Another UR ESI requestor received mixed methodological analytical support (power analysis for biospecimen sample size) on a SHERC pilot project with a clinical focus.

## 4. Discussion

We have presented the development process of a service center during the first five-year cycle of funding from NIMHD as an RCMI. The TAG-SC was created to serve the needs of university investigators who conducted health-equity related research. The team consisted of faculty and staff with expertise in research designs, methodologies, and analytic techniques, which helped to increase the institution’s research capacity. The initial outcomes from the development of the TAG-SC directly align with the goals of the RCMI Program Specialized Centers [5]. As previously mentioned, these goals include (1) fostering environments conducive to career enhancement with a particular emphasis on the development of new and early career investigators; (2) enhancing the quality of all scientific inquiry and promoting research on minority health and health disparities; and (3) enhancing institutional research capacity within the areas of basic Biomedical, Behavioral, and/or Clinical research. Results from the newly created TAG-SC suggest that the vast majority of research supported is Social/Behavioral and Clinical for ESIs and post-doctoral fellows on RCMI related projects.

A key driver for the conceptualization of the REDCap system centered around evaluation/tracking questions that enhanced SHERC’s ability to make a direct connection between the work conducted through the TAG-SC and the goals of the RCMI program. Key constructs that inform RCMI goals include users (ESIs and post-docs), understanding the specific types of methodological needs and how they shift through time (longitudinal analysis of the kinds of support requests), types of projects receiving support (Social/Behavioral, Biomedical, or Clinical), and timing of methodological consultation (proposal development or post-award research support). Using a REDCap ticketing system to support methodological consultations directly applies to RCMI institutions building methodological research capacity. Such institutions could adapt the SHERC TAG-SC structure to their specific needs (contact the corresponding author for access to the REDCap project template). Beyond RCMIs, this process for programmatic tracking needs can be implemented for intra- and inter-institutional collaborations. Ultimately, creating the REDCap system increased the teams’ efficiency in tracking information and created a centralized system with a digital record of all stages/processes, which increased the team’s ability to assist researchers.

### 4.1. Lessons Learned

In the following sections, we will discuss lessons learned and challenges faced during the development of the TAG-SC in hopes of helping other RCMIs if they choose to create a similar center.

#### 4.1.1. Improved Tracking and Evaluation

The challenges during Phase I (initial framework) underscored the need for a more robust tracking system. Programmatic synergy was critical to building a functional system that addressed all stakeholder’s needs. The SHERC Research Infrastructure and Administrative Core teams worked closely to develop a framework in REDCap that would best support the needs of the overall service center (reporting, billing, staffing, etc.), the TAG-SC team (documentation of consultation hours, accountability for level of service, and immediate tracking of ticket status), and the evaluation team (capture key metrics, support an ongoing feedback loop, and document outcomes). An iterative approach allowed the team to review each component of the system (request, assignment, etc.) and inform whether the proposed REDCap architecture would meet the needs of each stakeholder group. To consolidate the work, limit inaccurate reporting/poor recall, and support the evaluation team’s overall tracking of the RIC programmatic effort, the RIC coordinator worked with the evaluation team to create a data repository system for the RIC Faculty Assistance team members. As the RIC coordinator acquired new knowledge and REDCap added additional external modules, the ticketing system was improved.

#### 4.1.2. Limited Number of “Shoulder Taps”

Researchers with technical backgrounds become known at the university. Other researchers often seek “help” or expertise in another researcher’s field. While this increases interdisciplinary research efforts, these efforts are usually completed without compensation or acknowledgment. Hence, such requests for “help” often do not receive a response, mainly when the requestor and the helper are unacquainted, or the helper is overwhelmed by their own workload. TAG-SC provided a platform to improve this scenario for both the requestor and the helper. TAG-SC specialists can now direct these requests to the TAG-SC, where they are compensated for their efforts. Additionally, we can track the number of consultations and collaborations through the REDCap, allowing for better oversight of individual efforts. The faculty or staff with expertise is more likely to respond to the requestor, as they can track their time and contribution through TAG-SC, thus furthering the probability of collaboration. TAG-SC also provided structure and guidelines regarding the support being provided, which made it efficient to have discussions around authorship or acknowledgment, thus promoting efficient and equitable research practices. Finally, these requests can expand from one-on-one assistance to team assistance—furthering team science endeavors.

### 4.2. Limitations

As a data repository, REDCap has several automatic settings to provide data security, accuracy, and clarity. Some security settings (when implementing certain types of projects, like longitudinal studies) prevent certain types of database editing (i.e., Record ID numbering preferences). This can limit how the TAG-SC database is archived, stored, and utilized by various cores. For instance, grant and academic years do not align, creating overlapping datasets in archiving work for TAG-SC specialists, program evaluation, NIH Research Performance Progress Report (RPPR) reporting, and financial record keeping.

One of the most significant difficulties encountered following Phases I and II was the lack of uniform measures across both phases. Although changes to measures continuously improved the system, it created data integrity issues across phases in these early years. As with any data collection system, adding or removing fields impacts all records collected or to be collected. Over time, the many improvements and edits made to the ticketing system resulted in missing or removed data. In most cases, added fields can be retroactively filled in with administrative information (i.e., Grant Year, Academic Year, Level of Request). In contrast, others may be lost entirely (i.e., Date support requested, ESI status) if not recorded with the original request. The system has stabilized, allowing more reliable reporting as we move into the next funding cycle.

## 5. Conclusions

As part of the RCMI Program Specialized Centers, we were charged with (1) fostering environments conducive to career enhancement specifically for ESIs, (2) enhancing the quality of all scientific inquiry and promoting research on minority health and health disparities, and (3) improving institutional research capacity within the areas of basic Biomedical, Behavioral, and/or Clinical research. To directly achieve these goals, we have successfully developed the TAG-SC to support faculty engagement in health equity-related research at a southwestern RCMI institution. Through an iterative process, we have also developed a system to track programmatic functions that leads to an efficient and transparent data repository. There was a documented increase in faculty use of the TAC-SC through all phases of the program’s development, which was captured through the ease of accessibility of the REDCap tracking system. Specifically, in the first three years (2017–2020, Grant Years 1–3), 53 requests were made for support from RIC, with over half for proposal development involving Social/Behavioral research projects (Goal 3 of RCMI Program Specialized Centers). In the following two years (2020–2022, Grant Years 4–5), 86 tickets were submitted, a 40% increase over the previous period. This increase resulted from (1) the creation of the TAG-SC, (2) adding faculty and staff expertise, (3) the utilization of the REDCap ticketing system, and (4) marketing across the university to make researchers aware of available resources within the TAG-SC. During these two years, nearly a quarter of the tickets came from ESIs (Goal 1 of ECMI Program Specialized Centers), and over half requested assistance with funded projects or proposal development focused on minority health and health disparities research (Goal 2 of ECMI Program Specialized Centers). As in the first three-year period, over half of the tickets were for Social/Behavioral research projects (Goal 2 of ECMI Program Specialized Centers). Finally, TAG-SC creates an accessible university-based system for encouraging team science, enhancing and promoting the quality of health equity research, and expanding institutional research capacity by placing expertise at one’s fingertips.

## Figures and Tables

**Figure 1 ijerph-20-00191-f001:**
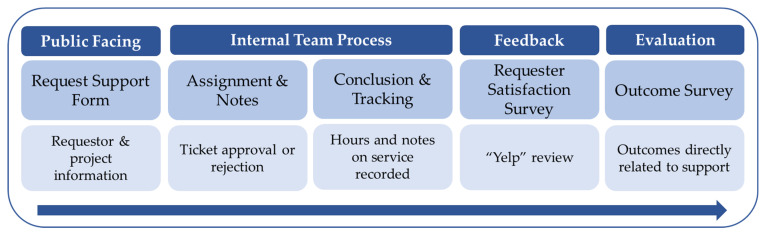
Time points in the REDCap project from the initial request to evaluation of outcomes.

**Figure 2 ijerph-20-00191-f002:**
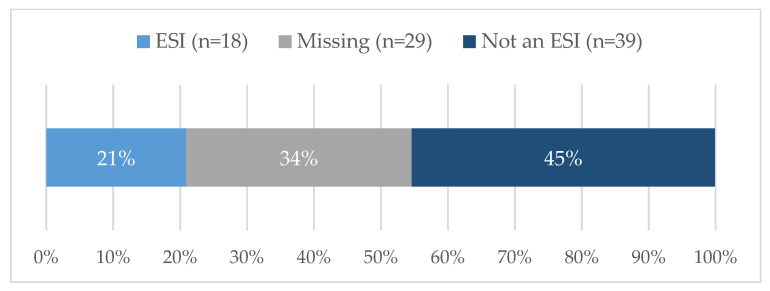
ESI TAG-SC user rates during Phase II: Pilot Year and Phase III: Year 1 of TAG-SC (N = 86).

**Figure 3 ijerph-20-00191-f003:**
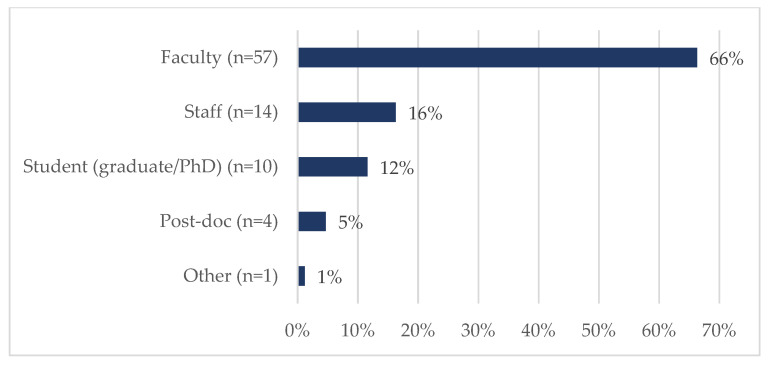
The professional role of TAG-SC users during Phase II: Pilot Year and Phase III: Year 1 of TAG-SC (N = 86).

**Figure 4 ijerph-20-00191-f004:**
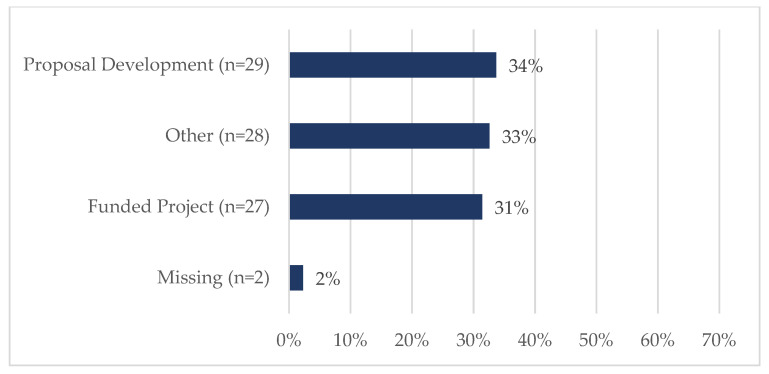
The project details associated with each request through the TAG-SC during Phase II: Pilot Year and Phase III: Year 1 of TAG-SC (N = 86).

**Figure 5 ijerph-20-00191-f005:**
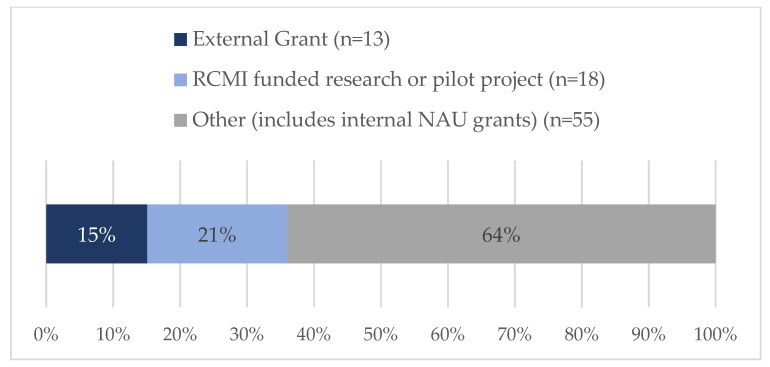
Project type associated with each request through the TAG-SC during Phase II: Pilot Year and Phase III: Year 1 of TAG-SC (N = 86).

**Figure 6 ijerph-20-00191-f006:**
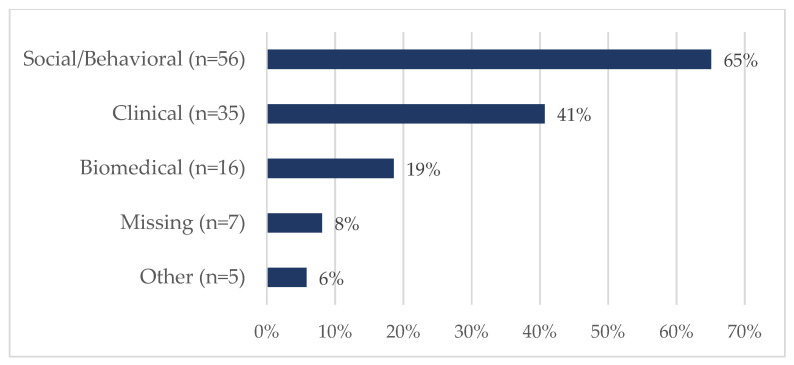
RCMI project content area for TAG-SC support (N = 86 *) during Phase II: Pilot Year and Phase III: Year 1 of TAG-SC. * Respondents could select more than one area. Thus the number of responses is greater than the number of respondents.

**Table 1 ijerph-20-00191-t001:** Example of questions included in Phase I of the REDCap ticketing system.

Question	Response Options
Professional role	Assistant Professor, Associate Professor, Professor, PhD student, Post-doc, Staff, Lecturer, Other
Type of support	Qualitative methodological/analytical support, Quantitative methodological/analytical support, Instrument development, Data management, Other
Research domain, if applicable	Social/Behavioral, Clinical, Biomedical, Biomedical & Clinical, Biomedical & Social/Behavioral, Social/Behavioral & Clinical
Type of project	SHERC pilot project, SHERC core project, SHERC campus community partnership, Other internal grant, External grant, Other
Project context	Funded project, Proposal development, Other

**Table 2 ijerph-20-00191-t002:** Variables showcasing types of requests during Phases II and III TAG-SC development.

Variable	Common Response Options
Early Stage Investigator (ESI) Status	Yes, ESI No, not an ESI
Professional role	Faculty Student (graduate/PhD) Post-doc Staff Other
Request type	Mixed methodological support Qualitative methodological/analytical support Quantitative methodological/analytical support Instrument development Data management REDCap assistance Other
Project content area	Social/Behavioral Clinical Biomedical Other
Project type	External grant SHERC funded research or pilot project Other (includes University internal grants)
Project details	Funded project Proposal development Other

## Data Availability

The de-identified data presented in this study are available on request from the corresponding author.

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
