# Peer review of "Building Research Infrastructure: The Development of a Technical Assistance Group-Service Center at an RCMI"

_ijerph, 2022, doi:10.3390/ijerph20010191_

Round 1

Reviewer 1 Report

Even the long-term work on the project described in the paper has to be appreciated, this paper seems to be more a report on the implementation of a specific funded project than a research paper to be published in a high visibility and quality journal. Therefore it does not lead to scientific conclusions, but only to a resume on the work performed.

Reviewer 2 Report

I enjoyed reading this article, thank you to all authors.

Can you please itemize the contributions from the beginning to the current stage? 

Reviewer 3 Report

The abstract must provide the reader a brief description about the whole paper within a concise paragraph. It can be formed as: a few-sentences introduction, followed by short highlights of the plot, procedure and the results. I believe the abstract can be improved.

Round 2

Reviewer 1 Report

This paper is in fact only a project report or at the very best it can be considered a general conference paper, but not a scientific article to be published as a scientific article in a high visibility journal.